# Toxic Effects of Inhaled Vanadium Attached to Particulate Matter: A Literature Review

**DOI:** 10.3390/ijerph18168457

**Published:** 2021-08-10

**Authors:** Marcela Rojas-Lemus, Nelly López-Valdez, Patricia Bizarro-Nevares, Adriana González-Villalva, Martha Ustarroz-Cano, Armando Zepeda-Rodríguez, Francisco Pasos-Nájera, Isabel García-Peláez, Norma Rivera-Fernández, Teresa I. Fortoul

**Affiliations:** 1Departamento de Biología Celular y Tisular, Facultad de Medicina, Universidad Nacional Autónoma de México (UNAM), Ciudad de México 04510, Mexico; marcelarojaslemus@hotmail.com (M.R.-L.); nelly_ciencias@hotmail.com (N.L.-V.); pbizarro@unam.mx (P.B.-N.); hemadgovi@yahoo.com.mx (A.G.-V.); ustarroz@unam.mx (M.U.-C.); armandzeped@me.com (A.Z.-R.); fcopasos@hotmail.com (F.P.-N.); igarciapelaez@yahoo.com (I.G.-P.); 2Departamento de Microbiología y Parasitología, Facultad de Medicina, Universidad Nacional Autónoma de México (UNAM), Ciudad de México 04510, Mexico; normariv@unam.mx

**Keywords:** suspended particulate matter, inhaled vanadium, toxicity

## Abstract

Environmental pollution is a worldwide problem recognized by the World Health Organization as a major health risk factor that affects low-, middle- and high-income countries. Suspended particulate matter is among the most dangerous pollutants, since it contains toxicologically relevant agents, such as metals, including vanadium. Vanadium is a transition metal that is emitted into the atmosphere especially by the burning of fossil fuels to which dwellers are exposed. The objective of this literature review is to describe the toxic effects of vanadium and its compounds when they enter the body by inhalation, based especially on the results of a murine experimental model that elucidates the systemic effects that vanadium has on living organisms. To achieve this goal, we reviewed 85 articles on the relevance of vanadium as a component of particulate matter and its toxic effects. Throughout several years of research with the murine experimental model, we have shown that this element generates adverse effects in all the systems evaluated, because it causes immunotoxicity, hematotoxicity, neurotoxicity, nephrotoxicity and reprotoxicity, among other noxious effects. The results with this experimental model add evidence of the effects generated by environmental pollutants and increase the body of evidence that can lead us to make more intelligent environmental decisions for the welfare of all living beings.

## 1. The Relevance of Environmental Pollution, Particulate Matter and Metals in Living Organisms’ Health Issues

The World Health Organization (WHO) recognizes that air pollution represents an environmental risk and a major health problem globally, affecting low-, middle- and high-income countries. Low- and middle-income countries are the most affected by environmental pollution problems and, therefore, develop health problems. It has been estimated that outdoor air pollution in 2016 caused 4.2 million premature deaths worldwide, from which 91% occurred in these countries [1].

Particulate matter (PM) is a common proxy indicator for air pollution and is estimated to affect more people than any other pollutant. PM is a complex mixture of solid and liquid particles, as well as organic and inorganic substances that are suspended in the air [1]. These particles are classified according to their aerodynamic diameter within the following categories:

Coarse particles: They have an aerodynamic diameter of less than 10 μÌm (PM_10_) and more than 2.5 μm (PM_2.5_) [2,3] and are mainly composed of silica, titanium, aluminum, sodium, iron and chlorides [3,4].

Fine particles: They have an aerodynamic diameter equal to or less than 2.5 μm (PM_2.5_) [2,3] and are composed of products derived from diesel combustion [5], such as carbon, lead, vanadium, sulfur dioxide and nitrogen dioxide [2].

Ultrafine particles: They have an aerodynamic diameter of less than 0.1 μm (PM_0.1_) [2,3] and are composed of the burning products of agricultural and forest, as well as gasoline and diesel combustion. However, they remain isolated in the atmosphere for short periods of time, because ultrafine particles tend to bind to larger particles [2].

PM with a diameter of 10 μm or less (≤PM_10_) can penetrate and accumulate deep within the lungs. The most harmful are the fine and ultrafine particles (PM_2.5_ and PM_0.1_) because they cross the blood–gas barrier and enter into the circulatory system [1]. There is a close and quantitative relationship between exposure to high concentrations of small particles (PM_10_ and PM_2.5_) and increased mortality and/or morbidity. Chronic exposure to PM increases the mortality risk. Small-particle pollution has an important impact on health even at very low concentrations; in fact, no threshold below which no harm to health is observed has been identified [1].

In 2013 the International Agency for Research on Cancer (IARC) determined that there is sufficient evidence to classify environmental pollution as carcinogenic to humans. In addition, the harmful potential of suspended particles, one of the most important components of environmental pollutants, was evaluated separately and classified as carcinogenic, both within Group I. Although the composition of air pollution and exposure levels may considerably vary from place to place, the IARC’s findings apply worldwide [6], therefore, addressing all risk factors for non-contagious diseases, including air pollution, is key to protect public health [1]. 

The importance of PM lies in the fact that they are receptors and holders of toxicologically relevant components of environmental pollution [2,5]. The particles generated by fossil fuel combustion and its derivatives are an example of this. These particles consist of a coal nucleus and the complex mixtures of compounds that are adhered to it, such as polyaromatic hydrocarbons, toxins, sulfates, nitrates and especially transition metals, such as vanadium, manganese, chromium, among others [7,8]. Metals are considered to play a very important role in the induction of toxic effects in biological systems [9,10].

Metals are the largest category of pollutants globally distributed. Their tendency to accumulate in some tissues of the human body, as well as their toxic potential at relatively low concentrations, is remarkable [11]. Constant exposure to certain metals has been linked to inflammation, cell damage and cancer, particularly of the lungs and skin [12]. Each metal has its own action mechanisms; while some are able to produce adverse effects by themselves, others interact with various components in the organisms [12]. 

It is important to mention that the presence of metals such as nickel and vanadium modify the physicochemical properties of PM, because when they are present, especially in PM2.5, it is observed that there is a positive correlation between the increase in mortality in the exposed population and the average concentrations of nickel or vanadium [13]. In another study conducted by Dominici et al. [14], with the same metals, similar results were found.

This review focuses on vanadium (V), one of the transition metals bound to particulate matter, because it is representative of the metal contaminants that contribute to health damage caused by PM. The objective of this review of the literature is to describe the toxic effects of inhaled V, especially the results of a murine experimental model that has served to elucidate the systemic effects that vanadium has on living organisms.

## 2. Vanadium

Vanadium is a transition metal member of group VB on the periodic table [15] that is widely distributed in the Earth’s crust, where it is present in quantities comparable to those of copper, nickel and zinc [16]. It is the 21st most abundant element [17]. It exists in oxidation states ranging from −1 to +5; the most common are +3, +4 and +5 [18]. It is not found as pure V (it is part of approximately 70 minerals), because it is very prone to react with other elements, especially oxygen, with which it forms V-oxides [18].

Due to its hardness, its ability to form alloys and its high resistance to corrosion, V is frequently part of the steel used to manufacture tools, orthopedic implants [19] and machinery [18]. Its use is common in the fabrication of semiconductors, photographic developers and coloring agents utilized to produce yellow pigments and ceramics. Salts, including vanadium oxides, are used as catalysts to produce sulfuric acid, in gas purification and in the oxidation of ethanol [18,20]. Vanadium pentoxide is a widely used commercial compound [19,21] and is the main raw material involved in the generation of other V compounds [22]; for example, from the reduction of V_2_O_5_, VO_2_ and V_2_O_3_ are obtained [21]. On the other hand, it is the most common compound in industrial exposure situations [21,22].

### 2.1. Emission Sources 

The total world emission of V is estimated to be between 71,000 and 210,000 tons per year [23]. Anthropogenic sources produce two-thirds of the V that is emitted into the atmosphere, specifically as oxides. Natural sources, such as continental dusts, marine aerosols and, to a lesser extent, volcanic emissions, produce the rest of the environmental V [18].

### 2.2. Routes of Exposure

The two main routes of exposure to V are the oral and the inhaled route.

Oral route: For the general population, food is the major source of V exposure [15,18,19]. The foods rich in V include black pepper, mushrooms, parsley [15] and spinach [18]. In general, foods of marine origin contain higher amounts of V compared to those of terrestrial origin [18,19]. However, the absorption of V-compounds in the gastrointestinal tract is poor [15]; only <1–2% of the metal that is ingested is absorbed [18]. 

Inhaled route: Vanadium is better absorbed and distributed in the body when inhaled [24]. It is considered that up to 90% of this metal is absorbed this way [25]. Inhalation is the main route of entry into the body [26]. Vanadium is adsorbed to the PM surface, especially to PM <2.5 μm, which easily enters into the alveolar sacs of the lungs. 

### 2.3. Toxicokinetic and Toxicodynamic

PM-bound V easily enters the respiratory tree penetrating deeply into the airways [27], crossing the blood–gas barrier and reaching the bloodstream [28]. The biologically relevant states of V are vanadate V (V), the pentavalent state usually as oxyanion (e.g., HVO42- or H2VO4-), a compound that predominates in the extracellular space; and vanadyl V (IV), the tetravalent state usually as oxycation (e.g., VO2+), which is more stable and predominates in the intracellular space [29]. In plasma, V is rapidly reduced to V (IV), generally by antioxidants in the plasma, and is transported bounded to proteins [30]. However, both oxidation states exist in plasma, because vanadyl undergoes spontaneous oxidation to vanadate [15].

Vanadyl is transported mainly by albumin and transferrin, and vanadate is transported only by transferrin; vanadium is transported throughout the body to organs and tissues bound to plasma proteins [15,31]. Approximately 90% of the V in plasma is found primarily as vanadyl bound to transferrin and albumin [18]. From the plasma, the distribution of V is fast (t ½-1 h); it accumulates mainly in bone, kidney, liver, spleen, lungs and testis [30]. Vanadyl as a free cation has limited ability to cross cell membranes, but vanadate (which predominates in the extracellular space, at neutral pH) is absorbed five times faster than vanadyl [15]. Vanadate enters into the cells by nonspecific ion channels and intracellularly is bioreduced to vanadyl [15,18]. This bioreduction is carried out by cellular glutathione, catecholamines, cysteine, NADH, NADHP and L-ascorbic acid [32] and subsequently binds to proteins [30]. Its reduction is regulated by the redox mechanisms of the cell that control the balance of V oxidation states [30].

Inside the cell, V binds to the cytosolic and nuclear fractions. A small portion (~1%) is found in free form [30]; furthermore, small amounts of V present in the cell might remain bound to lipidic molecules and are therefore not available to produce immediate toxicity [15]. The elimination of V from the body is biphasic, with a rapid (10–20 h) and a long terminal phase (40–50 days) [16,17]. V that is not absorbed is excreted via urine, which is the main route of elimination, and only a small amount (<10%) is excreted in the feces [18].

Vanadium can be highly toxic to humans and some animals, because it can affect important cellular functions, including the cell cycle, signaling pathways and cell survival mechanisms [33]. This element interferes with a large battery of enzyme-containing phosphates, such as ATPases, phosphatases, kinases [15,34], ribonucleases [15], DNA polymerases, peroxidases [34], ATP phosphohydrolases, adenylate cyclase, alkaline phosphatases and proteins for DNA synthesis [35]. On the other hand, the expression of several genes is affected by V, such as tumor necrosis factor-α (TNF-α), interleukin-8 (IL-8), activator protein 1 (AP-1), ras, c-raf, mitogenic gene activator protein (MAPK), p53, nuclear factors-kβ (NF-kβ) and others [15].

Vanadium also interacts with the cellular redox system, as it has an influence on reactive oxygen species and free radical production, markers of lipid peroxidation, enzymatic and non-enzymatic antioxidants; it also impacts the activity of antioxidant enzymes [36].

Vanadate is widely recognized as a structural and electronic analog of phosphate [37] (VO43- and PO43-) because of its similarity in protonation reactions [38,39]. This chemical similarity helps explain the effect of this metal on phosphorylation reactions in biological systems [40]. 

On the other hand, the importance of vanadium as a trace element is not yet established. It is believed that vanadium is important for the normal cell function and development, as it is present in all tissues involved in glucose homeostasis, lipid metabolism, antioxidant functions and as an immunomodulator; however, its role as an essential element has yet to be established in both humans and animals [41].

### 2.4. Exposure Limits

In recent years, vanadium concentration in the air has increased. In 2020, the World Health Organization [42] considered annual average values for urban areas to be in the range of 0.05–0.18 μg/m^3^, while maximum vanadium concentrations of up to 2 μg/m^3^ occurred in the most densely populated areas, during the coldest time of the year and in the late afternoon hours.

The National Institute for Occupational Safety and Health [43] and the American Conference of Governmental Industrial Hygienists [44] have determined that the occupational exposure limit to V gases and dusts per 8-h shift is 0.05 mg/m^3^.

## 3. Health Effects

The toxic effects of V and its compounds are related to the exposed organism: the dose, the route, the length of the exposure and the nature of the compound. Vanadium oxides may be the most toxic, coupled with the fact that the toxicity of these compounds increases according to their valence; thus, the pentavalent state is the most toxic [30]. 

Because vanadium is used as an alloying element in the production of steel and other materials, occupationally exposed personnel are susceptible to its toxic effects; acute and chronic poisonings have been described in workers engaged in the vanadium industry. The clinical symptoms that occur most frequently are those that reflect its irritant effects, especially in the respiratory tract [42].

Signs and symptoms that occur because of V exposure include irritation of the eyes and throat, green discoloration of the tongue, metallic taste in the mouth, cough, difficulty breathing, bronchitis, epistaxis, dyspnea, eczema, conjunctivitis, headache, dry mouth, dizziness, nervousness, insomnia and tremors [44].

### 3.1. Inhalation Experimental Murine Model

Environmental pollution is a growing problem worldwide, and PM is one of the most relevant toxicological components. Our working group has dedicated its efforts to understand the effects of V, one of the least mentioned metallic representatives, because it is an agent that is usually found attached to PM.

This research, based on the experimental model of inhalation in CD1 mice, recreates the exposure route by which this metal is best absorbed and to which populations living in highly polluted cities are exposed. The aim of this experimental murine model is to elucidate the mechanisms of action of V, as well as to approach the causal relationships between the effects found with the murine model and the pathologies generated by environmental pollutants.

In 1999, the inhalation model in mice was described for the first time, using lead acetate [45], and in 2004, the first results with the model were published, this time using vanadium [46]. In this experimental murine inhalation model, mainly male CD1 mice are used; however, CD1 females and Wistar rats have also been studied. The experimental animals are placed in an acrylic box connected to the ultranebulizer for 1 h, twice a week, for 1, 2, 4, 6, 8 or 12 weeks, according to the selected experimental protocol. The methodological details of each protocol are accessible in the corresponding publications from which the more relevant findings are summarized below. The advantage of the ultranebulizer (connected to an acrylic case) in this model is that it generates particles of 0.5–5 micrometers in diameter from a solution of 0.02 M of vanadium pentoxide, reaching a concentration inside the box of 2.57 or 1.56 mg/m^3^ if the solution is at 0.01 M, as represented in Figure 1.

### 3.2. Respiratory System

The respiratory system is one of the most susceptible systems to damage caused by air pollution, because it is the first site of contact with the environment. 

In our inhalation model, V triggered various alterations in both rat and mouse lungs. When rats were used as experimental animals, over a 6-week exposure time, V exposure produced thickening of the bronchiolar smooth muscle layer, non-ciliated cell bronchiolar epithelial hyperplasia and peribronchiolar inflammatory infiltrate. In addition, an increase in the pro-inflammatory cytokines IL-6 and TNF-α was observed in the bronchiolar epithelium (which may explain the afore mentioned morphological changes). On the other hand, in mice, 12-week exposure caused changes similar to those observed in rats [47,48].

In a tolerance model in which the mice were exposed to two inhalation phases with different V concentrations (in the first phase, 1.56 mg/m^3^, and in the second phase, 2.57 mg/m^3^), the following findings were observed: desquamation of non-ciliated bronchiolar cells, bronchiolar epithelial hyperplasia, small inflammatory foci and alteration in the expression of the CC16 protein. These changes occurred in both phases, but with less intensity in the second phase [49]. 

Overall, our evidence shows that V produces inflammatory changes in the lung, particularly in areas such as the bronchi and bronchioles, which could contribute to the development of hyper-responsiveness and responses similar to asthma and/or chronic bronchitis.

### 3.3. Immune System

The function of the immune system is to recognize self- and non-self-antigens in the organism, it is capable of being tolerant or mount an immune response. It is constituted by lymphoid organs, cells and molecules, which might be a target for air pollutants and therefore for V. 

*Spleen:* The spleen is a secondary lymphoid organ that has two essential functions: The first is to induce immune responses against soluble antigens and pathogens present in the blood circulation. This function is carried out in the white pulp. The second is to filter blood cell components (erythrocytes, platelets and leukocytes). This function is carried out in the red pulp. In the mouse, it is also a hematopoietic organ. In our work group, it was demonstrated that chronic inhalation of V induced macroscopic lesions in the spleen, which could cause histological and functional alterations [50]. Subsequently, to determine whether the observed histological lesions caused damage to the immune response, an immunohistochemical labeling was carried out for CD19 + (which is the marker for B lymphocytes). In addition, at the end of the exposure period (4 weeks), the mice were immunized intraperitoneally with the recombinant hepatitis B surface antigen (HBsAg), and the blood samples were used to identify the antibody against the recombinant surface antigen of Hepatitis B (HBsAb) by ELISA. The results showed that, in comparison with the mice of the control group, an increase in the number and size of the germinal centers was identified in the spleen of the exposed mice, and there was an increase in CD19 + cells. The average levels of HbsAb were lower than those of the control group. This indicates that chronic inhalation of V produces histological and functional changes in the spleen of exposed mice that induce dysfunction in the humoral immune response [51]. These findings are important for understanding the poor response of vaccinated children in cities with high amounts of air pollutants [52]. 

*Lymph nodes:* These are the secondary lymphoid organs responsible for filtering the lymph. The results obtained showed that in the lymph nodes of the exposed animals there was a significant increase in the number of CD25 + cells (B lymphocytes activation marker), especially in the medullary portion. The increase in germinal centers was also observed. These results reiterate that V causes an increase in B lymphocytes, which might be or not functional, affecting the humoral response [47,48].

*Thymus:* This is a primary lymphoid organ in which T lymphocytes from the red bone marrow complete their maturation and become immunocompetent. The thymus consists of two lobes, which, in turn, are subdivided into smaller lobules. In each lobule, the cortical and medullary regions can be identified. In addition to immature lymphocytes or thymocytes, there are several cell types in this organ: thymic epithelial cells (TECs), which are distributed in both the cortex (cortical thymic epithelial cells or cTEC) and the medulla (medullary thymic epithelial cells or mTEC); dendritic cells (DCs), which are distributed in the corticomedullary (cDC) and medullary (mDC) zones; and finally, macrophages, which are distributed throughout the thymus. Maturation of thymocytes to immunocompetent T lymphocytes depends on the cortical and medullary microenvironments established by cTECs, mTECs and DCs. The mTECs, together with the cDCs, are responsible for the establishment of central tolerance through negative selection of T lymphocytes by apoptotic elimination of lymphocytes with strong recognition of self-antigens, to prevent the formation of autoreactive clones. The cTEC cells positively select T lymphocytes that strongly recognize major histocompatibility complex (MHC) molecules. At the end of the maturation process, the T lymphocytes leaving the thymus will be a CD4+ helper or CD8+ cytotoxic [53,54].

*Thymic dendritic cells (DC):* The inhalation of V produces changes in the cortex–medullary relationship in the thymus in exposed mice [47]. These changes can affect both positive and negative selection and, therefore, the maturation process of T lymphocytes. One of the populations involved in negative selection is DC, to identify changes in the number of these cells, as well as changes in the expression of their markers CD11c and MHC-II (the latter being antigen presenting cells or APCs). The results obtained by immunohistochemistry, and that were later confirmed by flow cytometry, show that there are fewer CD11c + cells in the thymus of exposed mice. Flow cytometry also demonstrated that inhalation of V decreases the intensity of expression of CD11c and MHC-II markers from thymic dendritic cells. Therefore, inhalation of V generates a decrease in DC cells, as well as the expression of its markers CD11c + and MHCII +. A possible consequence of the negative selection of T lymphocytes may be that altered autoreactive clones could be released into the circulation [53]. 

*Medullary thymic epithelial cells:* mTEC cells are antigen-presenting cells and therefore express MHC-II. They are found distributed in the cortex (cTEC) and in the medulla (mTEC). Both types are functionally and phenotypically different. The mTECs, together with the DCs, are responsible for the negative selection of autoreactive clones; in addition, the mTECs regulate in situ the production of regulatory T lymphocytes. To explore the effect of V on mTECs, these cells were identified with the specific marker for cytokeratin 5 (K5). The results showed immunopositivity areas for K5 in cortical regions of the thymus. Densitometric analyses revealed an increase in the expression of this cytokeratin in the first week of exposure that decreased in weeks 2 and 4, but without returning to the initial expression levels [54].

*Nitrosative stress:* A probable cause of the changes in DC numbers, mTEC numbers and the morphological alteration in the thymus is an increase in nitrosative stress. To explore this possibility, the marker 3-nitrotyrosine (3-NT) was identified by immunohistochemistry in thymuses of exposed mice. The results showed that the 3-NT+ cells were distributed both in the cortex and in the medulla, while the densitometric analysis revealed that the expression of this marker increased significantly as the exposure time advanced. Confirmation of the increase in nitrosative stress may explain the observed changes in the thymus and partly explain the deterioration of the immune response in the exposed animals [55].

These changes strongly suggest that both negative and positive selection in which mTEC, DC and cTEC are involved may be altered, which would cause the release of autoreactive T lymphocytes into the circulation. These results support the hypothesis that atmospheric pollutants, especially PM, are capable of inducing changes in the systemic immune response, a fact that could explain the increase in a variety of autoimmune diseases, such as lupus or rheumatoid arthritis [56]. 

### 3.4. Hematopoietic Tissue

We reported that inhaled V affects the two cells in peripheral blood and some cells in hematopoietic and lymphoid organs. The most evident finding in peripheral blood was thrombocytosis, that is, the significant increase in the number of platelets in individuals exposed to V subacute inhalation (week 4), and which increased even more in subchronic exposure (week 8). Changes in platelet morphology were also reported, with the presence of gigantic platelets [50,57]. For this reason, the hematopoietic organs in mice, the bone marrow and the spleen were analyzed, in which an increase in the number and size was found for megakaryocytes, which are the cells from which platelets originate [58]. Subsequently, by means of an ultrastructural analysis, morphological differences were reported in the megakaryocytes of mice exposed to V, such as a greater number of granules and more lobed nuclei that indicate greater ploidy than in the control individuals [59].

One of the reasons that can explain megakaryocytosis and thrombocytosis may be the inflammation induced by V, since there are some cytokines, such as IL-6, that can increase the production of thrombopoietin in the liver. However, the mechanism that was considered, at least in part, responsible for megakaryocytosis and thrombocytosis in this model was the activation of the JAK/STAT (Janus Kinase/Signal Transducers and Activators of Transcription) pathway, since JAK2 and STAT 3 phosphorylated levels were elevated in subchronic exposed mice [60]. This pathway activates the proliferation and antiapoptotic mechanisms of megakaryocytes. Our possible explanation is that the increase in phosphorylation can be explained by the well-known effect of V as an inhibitor of protein tyrosine phosphatases (PTP), enzymes important in the regulation and inhibition of signaling pathways [61].

Our working group also wondered if these elevated platelets were functional. To answer this question, platelet aggregation tests were performed, and it was found that the aggregation decreases at week 4 of exposure. However, at week 8, normal platelet aggregation was observed. We corroborated this inhibitory effect on platelet aggregation due to V by conducting an in vitro study with different concentrations of V [62]. It is interesting to note that platelet activation markers, such as P-selectin, were elevated two weeks after the exposure was suspended. In the model, no alterations in hemostasis were observed; however, we consider that it is convenient to study a longer V exposure, due to the fact that there are factors in this model that could predispose individuals to generate a thrombotic phenomenon, such as an increase in activation markers (e.g., the increase in plasma fibrinogen) [57]. Moreover, there is evidence that supports that V is capable of causing endothelial damage, as was explored in an in vitro study [48,63]. 

In addition, a decrease in hemoglobin and hematocrit was found. The exposed individuals presented normochromic normocytic anemia in subchronic and chronic inhalation exposure (weeks 8 and 12). We have not yet determined the mechanism of V toxicity on erythrocytes; however, other authors have proposed hemolysis due to osmotic fragility caused by the inhibition of the Na/K ATPase pump or by cell death known as eryptosis, both induced by V [64,65]. However, other studies are necessary to determine if anemia is caused by hemolysis or by a decrease in erythropoiesis [48]. Leukocytes are also affected by V inhalation. Leukocytosis with neutrophilia and lymphocytosis were observed in the blood samples [57]. These cells may be elevated in the blood because of the acute and chronic inflammation that V induces initially in the lung, and later in other organs. Morphological alterations were also observed in lymphocytes, some of which were binucleated [57]. In other studies, nuclear changes were also reported, such as invaginations and pseudo-inclusions, which may be caused by alterations in the cytoskeleton induced by V, such as the inhibition of the polymerization of both actin and tubulin [47,66,67].

The genotoxic effects of inhaled vanadium have also been studied by using the micronuclei (MN) technique stained with acridine orange, in which micronuclei are quantified by reticulocytes. An increase in MN frequency is considered to be strong evidence of the genotoxic effect of an agent. When both female and male mice were exposed for 4 weeks, it was observed that, in both cases, the frequency of MN increased significantly, and when sex comparisons were made, it was shown that males are more susceptible to DNA damage caused by this metal. In peripheral blood leukocytes from the same experimental animals, V did not cause cell death, indicating that it did not cause cytotoxic effects. Hence, the results suggest that V generates damage to genetic material without causing cell death, thus increasing the probability of developing pathologies such as cancer [68].

### 3.5. Nervous System

The results obtained with the inhalation model show that V induces cytological, ultrastructural and functional alterations in different structures of the central nervous system.

In the olfactory bulb, exposure causes a decrease in dendritic spines in granule cells, in addition to inducing apoptotic and necrotic cell death of these neurons [69]. The increase in the activity of glutathione peroxidase (GPx) and glutathione reductase (GR), the reduction in olfactory function [70] and the increase in the levels of the oxidative stress markers were confirmed by malondialdehyde (MDA) in the same structure [71]. These results show that V alters the structure of granule cells, inducing their death by oxidative stress, which directly interferes with the olfactory function.

In the cerebral ventricles, alterations were found in the ependymal epithelium, such as the loss of cilia, the compaction of the remaining cilia in the epithelium, the desquamation of the cells due to the loss of intercellular junctions and the generalized disintegration of the histological structure [72]. Subsequently, the generation of oxidative stress was confirmed through the lipid peroxidation marker 4-hydroxynonenal (4HNE) in the choroid plexuses [48]. Together, these results show that V can alter the blood–brain barrier, making the system more susceptible to damage; this helps to explain the structural damage that was observed.

Other of our findings demonstrated that V induces the activation of MMP-9 and MMP-2 gelatinases in the olfactory bulb, prefrontal cortex, hippocampus and striatum [73]. This activation could be related to the prior-mentioned alteration of the blood–brain barrier and to changes, such as loss of spines and neuronal death, in the mentioned structures. The loss of dopaminergic neurons in the substantia nigra and the subsequent loss of spines in medium-sized spinous neurons in the striatum has also been demonstrated [72]. These structures are especially vulnerable because of its low antioxidant concentration, making them more susceptible to the oxidative damage produced by V.

V inhalation also produced loss of dendritic spines in pyramidal neurons in the CA1 region on the hippocampus and in the neuropil in the same region. In addition, cell death was observed due to necrosis of the granular neurons of the dentate gyrus and the deterioration of spatial memory [74]. The alteration of spatial memory due to neuronal damage is known to be a consequence of aging and neurogenerative diseases; therefore, this is a metal that could be related to the development of this type of pathology. 

Finally, other evidence shows that inhalation of V produces damage to the retina detected through the damage markers GFAP (Glial Fibrillary acid protein) and GS (glutamine synthase). In this study, the decrease in GFAP in Müller cells was demonstrated at 8 weeks of exposure, which suggests dedifferentiation of the glia. In addition, the decrease in glutamine synthase was detected in a manner dependent to the exposure time, which suggests the degeneration of the photoreceptors [75]. These findings show that V can alter the structure and, most likely, the function of the retina and might interfere with vision.

### 3.6. Urinary System

V affects the urinary system, mainly the kidneys. In our model, it was observed that vanadium inhalation (week 6) causes an increase in the size of the kidneys, as well as swelling of the cortex. From week 4 of exposure, the expansion of intraglomerular mesangial cells is observed, as well as the thickening of the muscular layer of the arterioles [76]. In a later study, we report that, in subchronic exposure (week 8), there are alterations in the morphology of tubular cells. Among them, the following can be observed: decrease in microvilli, alteration of cell shape (since some had protrusions), vacuolization and detachment of tubular cells. The lumen of the renal tubules was occupied by an eosinophilic material. Inflammatory foci and increased oxidative stress were also observed. This finding was identified by immunohistochemistry of the increase in 4-hydroxynonenal, a marker of lipid peroxidation. In addition, these changes were correlated with changes in the renal function determined in the urine, where proteinuria and hematuria were reported [77].

### 3.7. Male and Female Reproductive Systems

The reprotoxic effect of vanadium has been reported in animal models, both in males and females [47]. 

*Males:* In the case of males, vanadium accumulates in the testes from the first exposures to the inhalation of this metal [66], thus making testicular cells susceptible to the toxic effect of V. Spermatogenic cells have been shown to be targets for vanadium toxicity, including cell death from necrosis and apoptosis; intraperitoneal administration of vanadium tetraoxide causes increased apoptosis in germ cells in stages I–III and XII of the seminiferous epithelium cycle [78], while inhalation of V causes necrosis of spermatogonia, spermatocytes and Sertoli cells [79]. Alterations in the ultrastructure of testicular cells have also been observed, such as cellular inclusions in germ and Sertoli cells [78], as well as nuclear pseudo-inclusions in germ cells [79]. Vanadium can directly cause damage to testicular cells, as reported by Aragon and collaborators [78], by finding apoptosis and cell inclusions, with the serum testosterone values unaffected by intraperitoneal administration of the tetroxide of vanadium, indicating that vanadium is likely to damage the blood–testicular barrier. This barrier is vital for the development of germ cells, and it has been observed, with the use of electron microscopy, that the inhalation of V causes an interruption of the intercellular junctions in the seminiferous tubules [79]. Spermatogenesis is a synchronous mechanism of cell differentiation which involves communication between Sertoli cells and germ cells, as well as between germ cells themselves; communicating junctions play a predominant role in intercellular communication, and we found alterations in the presence and distribution of connexin 43 (Cx43) in the seminiferous tubules because of inhalation of V. By immunohistochemistry, it was noticed that, not only does the presence of Cx43 decreases in the cytoplasm, but the presence of this connexin in the plasma membrane is also reduced [80]. The cytoskeleton also plays a predominant role in the regulation of spermatogenesis; actin filaments participate in the morphological changes that cells present during their development, in addition to being included as part of the intercellular junctions. In our experimental model, we observed that, from the third week of V-exposure, the presence of actin in testicular cells decreased, and this effect was progressively exacerbated until week 12 [67]. Microtubules are essential structures during cell division, in mitosis or meiosis, together with their participation in the transport of organelles and vesicles; inhalation of vanadium causes a decrease in tubulin gamma in germ cells, Sertoli cells and Leydig cells from the first week of inhalation, and this effect is accentuated depending on the exposure time [66].

*Females:* The toxic effect of V also affects females when they are exposed to the inhalation of this metal. In non-pregnant adult females, we have observed alterations in the estrous cycle, modifications in serum concentrations of sex hormones and histological changes in the ovary. Regarding the estrous cycle, up to 72.3% of the females that inhaled vanadium presented anestrus in the fourth week of exposure. In exposed females, low concentrations of 17β-estradiol and progesterone were determined; however, there was no difference in the presence of the estrogen receptor α in the uterus between the group of exposed females and controls. Regarding the histological alterations, the decrease in the diameter of secondary and pre-ovulatory follicles in the ovary and the decrease in the thickness of the myometrium and uterine stroma in exposed females were evident [81]. 

Vanadium is an environmental pollutant that is generated mainly by the burning of fossil fuels; its reprotoxic role has been demonstrated in animal models and in part of suspended particles (PMs), especially those that can enter the lung alveoli (PM_2.5_). Hence, it is important to point out this metal’s reprotoxic capacity, which can be associated with reproductive problems or infertility.

Figure 2 summarizes the most important findings and health implications that have been reported in the respiratory, urinary, reproductive and nervous systems, as well as in the hematopoietic tissue and in the liver (see Section 3.9 below).

### 3.8. Cardiovascular System

Our working group has conducted some research—though, not exhaustively—using organs of the cardiovascular system. The relationship between atmospheric pollution and the increased probability of suffering arrhythmias has been observed; however, the biological mechanism associated with these events is still unknown. One possibility is that the decrease in N-cadherin protein alters the disposition of connexin-43 protein (essential for the gap junctions present in myocardiocytes) and that this alteration predisposes subjects to heart-rate problems. Therefore, we analyzed myocardial histology and the expression of N-cadherin and connexin-43 by immunohistochemistry in CD1 mice exposed to vanadium inhalation. Our results showed a significant and progressive reduction of both N-cadherin and connexin-43, as well as the presence of meganuclei; disruption of myofibrils and their agglutination were also observed in the exposed groups. Our findings add a little more information about a possible explanation for the arrhythmogenic effect observed in the inhabitants of cities with high PM air pollution [82].

### 3.9. Glands of the Digestive System

One of the systems that has received less attention, when it comes to inhaled pollutants, is the digestive system. 

*Liver:* The liver is a very important organ due to its multiple functions: metabolism, in general; protein synthesis; and detoxification. Inhalation of V can affect the morphology and function of the liver, as has been demonstrated by histological alterations, such as chronic inflammation and steatosis, identified in the exposed mice. Because the liver is an organ that has a great repair capacity, some morphological manifestations have been found that indicate this process, such as megalocytosis, the presence of mega-nuclei, with nuclei of approximately 20–30 µm (controls between 7 and 10 µm) and a greater number of binucleated or even trinucleated hepatocytes. Damage to hepatocytes was also reflected in liver function tests, in which the enzymes alanine and aspartate transaminases (ALT and AST) were found elevated in the serum of mice at 6 weeks of V exposure [83]. One of the most studied mechanisms of V toxicity is the generation of oxidative stress, which we corroborated in the liver by measuring TBARS (thiobarbituric acid reactive substances), which were elevated at 6 weeks of exposure [48,83], and an increase in another marker of oxidative stress has also been found, which is 4-hydroxynonenal determined by immunohistochemistry [47]. Nowadays, an increase in the incidence of liver diseases has been observed, mainly non-alcoholic liver disease, where inflammation and steatosis occur and can have metabolic consequences. This disease is the consequence of various factors to which the inhalation of PM can be added.

*Pancreas:* There are very few studies that report pancreas damage by metals. In our model, it was found that, at 6 weeks of exposure, acinar cells showed increased cell volume and an increase in the number of granules. This increase in granules could be due to the inhibition of microtubule polymerization and the inhibition of secretion. These changes were also accompanied by serum pancreatic damage markers, such as increased pancreatic amylase, that could also indicate cell damage, highly possible because of lysosomal membrane damage induced by oxidative stress [48,84]. It would be interesting to continue with the study of pancreatic damage to observe if the endocrine pancreas and its hormonal secretion is also damaged. In addition, it would be interesting to determine its possible effect on the metabolism, because V is known to be an insulin mimetic agent. 

### 3.10. Inhalation of V and Consumption of Sweet Beverages

Air pollution is a condition that has recently been associated with a higher risk of obesity and metabolic disorders. It has been reported by several research groups that metals alter glycemic regulation [60]. In addition, people living in polluted environments change their lifestyle and may include dietary modifications that involve the consumption of sweet beverages. The combination of both factors (inhalation of vanadium and consumption of sweet beverages) is being studied by our group to identify the possible association or the increase in damage on different organs. In the case of the kidney, sweet beverages by themselves caused some damage; however, the combination of inhalation of V and the consumption of these beverages increased the observed damage, which also affected kidney function, finding a greater number of inflammatory foci; more severe changes in cellular damage, such as those mentioned for V inhalation; and the presence of significantly more marked proteinuria and hematuria in individuals exposed to the combination of both factors. The damage mechanism studied in this model was oxidative stress, showing that 4-hydroxynonenal increased in the V and sucrose groups, and this increase was more evident with the joined exposure [77]. 

The relevance of this finding is that these factors, metal pollution and sweetened beverages consumption, could partly explain the increase in recent years of the incidence of chronic kidney failure, for which a direct cause is not known. 

## 4. Conclusions

Exposure to environmental pollutants, especially PM, constitutes a health risk for living organisms. In this review, through the results obtained in an experimental mice model, we exposed the toxicological potential of V, one of the several metals bound to PM, because its participation as an environmental pollutant is scarcely mentioned.

The results in the inhalation model reported here demonstrated that vanadium generates histological and physiological changes in different cells and organs, including the lung, the lymphoid organs and the immune system, the hematopoietic tissue, the renal system and the central nervous system. In addition, results show that oxidative and nitrosative stress play a relevant role in the V toxic mechanisms; furthermore, cancer, infertility and metabolic diseases could be the result of PM exposure and, therefore, exposure to V. 

## Figures and Tables

**Figure 1 ijerph-18-08457-f001:**
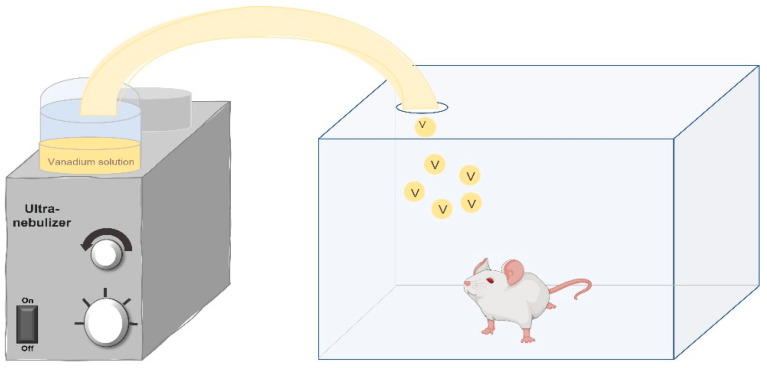
Schematic representation of the murine inhalation exposure model. Experimental animals are placed inside a transparent acrylic box which is connected to the ultranebulizer. The concentration of vanadium particles that reach the inside of the acrylic box is determined by the molarity of the vanadium solution. Original image created by the authors.

**Figure 2 ijerph-18-08457-f002:**
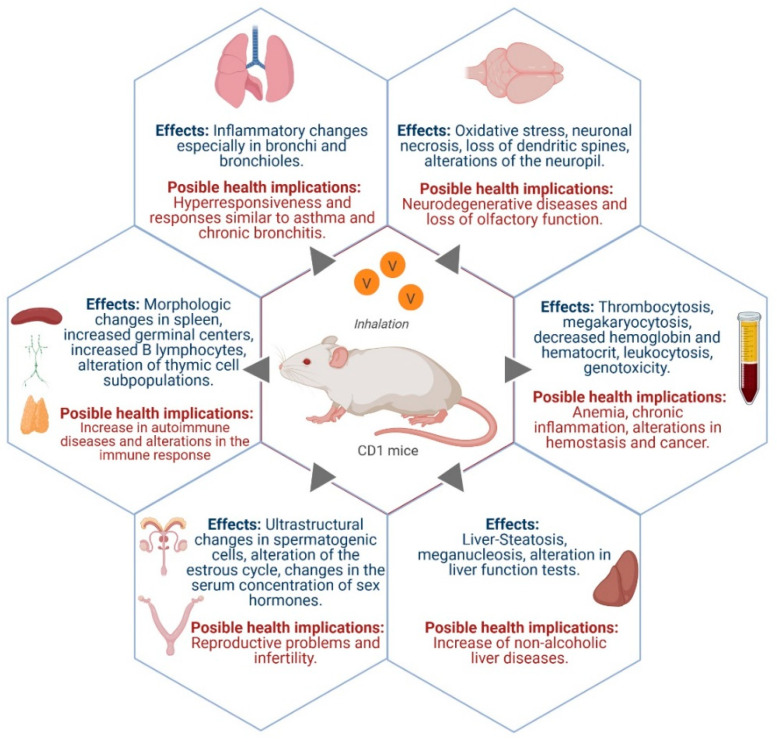
Some effects caused by vanadium inhalation in a mice experimental model and its possible health implications. Figure created with biorender.com.

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
