# Peer review of "Toxic Effects of Inhaled Vanadium Attached to Particulate Matter: A Literature Review"

_ijerph, 2021, doi:10.3390/ijerph18168457_

Round 1

Reviewer 1 Report

The manuscript has been improved and now it can be accepted for
publication in the IJERPH.

Reviewer 2 Report

Through the author's modification, the significance of the article has been highlighted, and the references have been updated. I think the review can be accepted.

This manuscript is a resubmission of an earlier submission. The following is a list of the peer review reports and author responses from that submission.

Round 1

Reviewer 1 Report

The author’s purpose of the investigation is very interesting, also for scientists from related research fields. I would recommend the suggestions described below:

  • The title should be the mirror of the paper. What is really new in the paper?
  • Abstract should referred to all the forms of vanadium particles as weel the other vanadium compounds, such as vanadate, vanadyl, etc. After reading the paper some informations is missing in the abs. The abs should reflect of the paper and not a kind of intro, aims or approaches.
  • The paper contains only about 9% of references in the last 4 years, not including the world organization reports within the topic. Therefore, references should be update, described and compare with the old ones. Comparison is a step forward in science.
  • In the introduction it is not clear the relevance of vanadium apllications. Moreover, if the authors felt adequate recent references to others vanadium application could be mention, for instance biological and biomedical apllications, as weel as toxic effects. That would turn the attention of the paper more wide-ranging.
  • At the end of the introduction it also not clear what is the main message and relevant points of the review paper that should be emphasize at this stage.
  • I would like to make the following suggestion: a figure or scheme with the chronology or a timelime of the major events in the field would be interesting and usual for a better understanding of the paper. Moreover, this timeline will also reflect the understanding of the authors about these events. This personal view timeline will be interesting and also pedagogical for the others researchers about the vanadium toxicity.
  • The Figure 1 is nice but no information is giving which references described the effects referred.
  • What the authors mean about others vanadium compounds in ref 18? Although This reference is obviously relevant it is from 2004. This could be update also.
  • In fact, for instance in section 2.3 references are from the beggining of the century or even from the XX century. Recent papers and review papers from expertise in vanadium field could eventually be added such a from the research goups of Agnieszka Scibior, Enrique Gonzalez-Vergara, JC Pessoa, E Garriba, Aureliano and/or Crans vanadium recent studies within several of the topics described in this section, among others recent papers and/or emerging topics in the field of vanadium toxic effects.
  • Also vanadium toxicological studies in humans should be emphasized and resume.
  • Finally, I would like also to propose to include a figure of the scheme of the experimental model, once it seems to be the origin, at least in part, of the review.
  • Globally the conclusions should followed the order of presentation of the paper with partial conclusions first and then global conclusions. Care must be taken not to speculate the overall application of the findings without facts in human models.

Reviewer 2 Report

The manuscript (MS) “Inhaled vanadium and its toxic effects”, contributes to environmental toxicology of vanadium. The topic fits the aims and scope of the International Journal of Environmental Research and Public Health.

In my point of view, MS needs in some transformation. Here is a list of corrections to be made to the text.

1) I recommend specifying in the title that this article is a review, and  that the review aimed  to vecotoxicology of vanadium in the composition of PM.

2)  It should be written in an abstract that the article is a literature review. It is necessary to write the aim of the review and the number of articles analyzed in review.

3) It is not entirely clear what the authors mean by the experimental model . Is this a mathematical model? Where is the description of the model ?

4) If the first point of the review is an Introduction, it needs to add in it  information about vanadium, as currently it  contains  only information on the hazards of PM air pollution.

5) I recommend to remove Figure 1 from the Conclusions. It must be transferred to the body of the review.

.

Reviewer 3 Report

The document contains an adequate and comprehensive review on the toxic effects of Vanadium. This article is very interesting because, normally, attention tends to focus on other toxic metals such as Al, Pb, Hg, etc. However, these types of pollutants are present in the natural environment and have been shown to be toxic.
My opinion is that this article should be published in this form.

Reviewer 4 Report

Vanadium is one of the transition metals widely present in the Earth's crust. It is often emitted into the atmosphere by the burning fossil fuels to which dwellers are exposed, and it is usually present in the form of binded to particulate matter. This review is very interesting because in literature there are hardly studies that well summarized the systemic health effects of this element on living organisms, although is representative of the metal pollutants that contribute to health damage caused by particulate matter. As such the review deserves publication in IJERPH.

Reviewer 5 Report

Multiple factors are involved in the pathogenesis of non-communicable diseases resulting from exposure to PM. This makes causal links between health effects caused by PM and PM exposure and properties (composition, size) challenging to establish, especially for chronic effects of long-term exposures to low PM concentrations. Vanadium as one of transition metals ofen bind to PM has an important influence on oxidative activity of PM. This manuscript systematacially introduced inhaled vanadium and its toxic effects, which plays a positive role in establishing the health hazards of transition metals in particulate matter. It is suggetsted that the authors can highlight the importance and significance of this review from the effects of vanadium on the physicochemical properties of PM (OPm, OPv, toxicity, etc,.).